# Examining the Decoupling of Economic Growth with Land Expansion and Carbon Emissions in Zhejiang Province, China

Zepan Li [1], Zhangwei Lu [1],*, Lihua Xu [1], Yijun Shi [1], Qiwei Ma [1], Yaqi Wu [1], Yu Cao [2] and Boyuan Sheng [1]

[1]  School of Landscape Architecture, Zhejiang Agriculture and Forestry University, Hangzhou 311000, China; lizepan@stu.zafu.edu.cn (Z.L.); xulihua@zafu.edu.cn (L.X.); yijun_shi@zafu.edu.cn (Y.S.); maqiwei@zafu.edu.cn (Q.M.); 20030050@zafu.edu.cn (Y.W.); staceysheng@hotmail.com (B.S.)
[2]  School of Economics and Management, Southeast University, Nanjing 211189, China; caoyu@seu.edu.cn
*  Correspondence: zhwlu@zafu.edu.cn; Tel.: +86-135-7546-8853

**Abstract:** Coordinating the interrelationships among economic growth, land resource utilization, and carbon emissions is critical for promoting high-quality economic growth and achieving sustainable urban progress. According to the gravity model and the Tapio decoupling model, this study examined the decoupling relationships of economic growth with land expansion and carbon emissions in Zhejiang Province during the period of 2002–2017. We found that (1) The economic gravity center and the built-up area gravity center generally shifted towards the northwest; however, the carbon emission gravity center initially shifted towards the northwest and then towards the southwest. The spatial coupling between the economic gravity center and the built-up area gravity center exhibited a tendency of 'first weakening, then strengthening, and last weakening', whereas the spatial coupling between the economic gravity center and the carbon emission gravity center displayed a tendency of 'first strengthening and then weakening'. (2) The decoupling of economic growth and land expansion is weak at every stage with effective controls on land expansion. However, in recent years, the phenomenon of 'expansive negative decoupling' has become prevalent in counties and cities surrounding the central city. The decoupling of economic growth and carbon emissions steadily increased at each stage, first 'expansive coupling and expansive negative decoupling', then 'weak decoupling', and finally 'strong decoupling'. The urban low-carbon transformation effect is remarkable. (3) Zhejiang Province should prioritize addressing the regional imbalance and state instability in the decoupling relationships. It is crucial to comprehensively consider the natural resource endowment, macro-policy factors, and urban development orientation of counties and cities while implementing differentiated planning and control strategies, which will promote regional coordination and comprehensive, high-quality development in all areas.

**Keywords:** economic growth; land expansion; carbon emissions; migration of gravity centers; decoupling relationship; Zhejiang Province

## 1. Introduction

### 1.1. Background

Coordinating resources, environmental development, and economic growth in the face of increasing resource constraints and deteriorating environmental conditions is a crucial scientific problem that urgently requires resolution [1,2]. Land resources, as the fundamental resource and spatial carrier for economic activities and urban construction, have played an essential role in promoting economic growth to a certain extent [3]. However, due to the path dependence of economic growth overly relying on land expansion, problems such as inefficient use of land resources and destruction of arable land resources have become increasingly prominent [4,5]. During the pursuit of economicgrowth, the expansion of industries that heavily rely on energy has resulted in a notable increase in both energy consumption and carbon emissions, exerting a detrimental influence on the progress of

sustainable urban development [6,7]. Currently, China's urbanization is undergoing a transformation from a speed-oriented model to a high-quality development model and is on the verge of entering a key stage of maturity and standardization [8,9]. Therefore, examining the decoupling of economic growth, land expansion, and carbon emissions is significantly valuable in assessing the quality of economic development. This endeavor is conducive to guiding economic transformation and promoting the construction of an ecological civilization.

Zhejiang Province was the first province to propose a new urbanization strategy and is at the forefront of China's new urbanization development. Despite this, the Province faces the challenge of limited land resources and high energy consumption. Given the increasingly severe constraints on resources and the environment, Zhejiang Province has proactively advocated for the establishment of a model province for land and resource conservation and intensification and has taken the lead in promoting ecological civilization construction. Therefore, this study examined the decoupling of economic growth with land expansion and carbon emissions in Zhejiang Province in the period of 2002–2017 at county levels. We applied the gravity center model to reveal the evolutionary trajectory and uneven distributions of economic growth, land expansion, and carbon emissions. We also used the Tapio decoupling model to explore the evolving trends and spatial patterns of their decoupling relationships. The findings provide references for further promoting high-quality development of counties and cities within the region, as well as for advancing the coordinated developments among the resources, environmental development, and economic growth.

### 1.2. Literature Review

The connections among economic growth, land expansion, and carbon emissions have long been subjects of academic interest. Multiple methods have been employed to explore the relationships among these factors, such as evaluating the inverted U-shaped relationship between two variables using the Environmental Kuznets Curve [10–12], analyzing the unilateral and dual causality relationship between two variables through the Granger causality test [13–16], and measuring the decoupling relationship between two variables by employing the decoupling model [17–19]. The EKC curve mainly reflects the long-term trend [20,21], and the Granger causality test can only determine whether two variables have a statistically significant causal relationship [22–24]. In contrast, the decoupling model can effectively identify specific stages in the evolution of a relationship through a simple quantitative relationship analysis [25,26], which is suitable for dynamic and multi-scale evaluations [27–29], and has become an effective evaluation method for measuring regional economy development [30,31]. The Organization for Economic Cooperation and Development (OECD) first introduced the concept of decoupling into the field of environmental economics to measure the synchronous relationship between economic growth and resource and environmental issues [32]. Subsequently, with the introduction of elasticity coefficients by Tapio [33], the decoupling status was further subdivided, becoming a main approach for studying the decoupling of economic growth from resource and environmental pressures.

Urban land expansion affects sustainable development and is closely related to social and economic growth. The rapid expansion of urban land has led to serious social and environmental issues, such as the reduction of croplands [34] and irreversible damage to natural habitats and biodiversity [35,36]. Numerous scholars have extensively explored the social, economic, and environmental consequences of urban land expansion from a global perspective [37,38]. Wei et al. [39] focused on cities within the "Belt and Road" regions experiencing rapid urbanization. Their investigation delves into the evolutionary patterns of urban expansion, adopting a perspective grounded in global land optimization management and sustainable development. Investigating how to achieve a strong decoupling relationship between economic growth and land resources is of great significance in reducing the depletion of natural capital at the land-use level [40].

In the context of China's economic transformation, the extensive and inefficient land use cannot meet the demands of China's economic development under the new normal. Therefore, new strategies are needed to formulate land use patterns and policies that align with China's strategic transformation [41]. Some scholars focus on exploring the social and economic factors behind urban land expansion to provide a scientific basis for guiding the transformation of land utilization patterns. Wu et al. [42] examines urban land efficiency in the Yangtze River Delta, a rapidly urbanizing region in China, focusing on spatial patterns and determinants related to accessibility and economic transition, highlighting the importance of urban land efficiency for sustainable development in the context of global urbanization and limited land resources. Other scholars have concentrated on coordinating the relationship between economic development and urban land expansion. For example, Yu et al. [4] examined the decoupling relationship between land use and socioeconomic development in 12 urban agglomerations comprising 184 cities, assessing land use efficiency and identifying driving factors.

The decoupling theory has been widely used to study the relationship between economic growth and carbon emissions. Some scholars emphasize the global goal of low-carbon economic development. Shuai et al. [43] analyzed economic growth and carbon emissions decoupling in 133 countries and found that income level significantly affects absolute decoupling, with higher-income countries achieving higher proportions of decoupling. Similarly, Wang et al. [44] investigated decoupling trends between economic development and carbon emissions in 192 countries and found that developed countries have shifted from a stable weak decoupling to a strong decoupling state driven by declining energy intensity, whereas most developing countries do not show significant decoupling at the stage where economic growth is the main priority. Meanwhile, emerging economies have become an important source of global carbon dioxide emissions, drawing the attention of scholars. Ozturk et al. [45] examined $CO_2$ emissions decoupling in Pakistan, India, and China from 1990 to 2014, finding successful decoupling in some years, but challenges were also encountered, such as negative decoupling in Pakistan and weak decoupling with high costs in India. Naseem et al. [46] investigated environmental quality and economic growth in BRICS countries, revealing a long-term interrelation between economic expansion and environmental degradation, underscoring the importance of sustainable policies and the application of green technologies.

As the world's largest carbon emitter, China faces increasing challenges in mitigating carbon emissions. Currently, research on the decoupling relationship between economic growth and carbon emissions in China's provinces and cities is relatively mature. Zhao et al. [47] found significant differences in the decoupling indices of economic growth and carbon emissions among different economic regions in China, revealing imbalanced development among provinces within each region. Other research also emphasized the need to recognize the diverse developmental characteristics in different cities in terms of economic, social, and environmental aspects in order to formulate targeted environmental regulations and policies to promote carbon reduction [48–50].

While decoupling relationships have been extensively studied at the national and provincial levels, there has been relatively less attention given to the county-level analysis. Examining the decoupling of economic growth with land expansion and carbon emissions at the county level is necessary to tailor differentiated land protection and carbon reduction strategies in accordance with local conditions. This study holds theoretical and practical significance in enhancing the new urban development strategy, promoting high-quality economic transformation, and narrowing the gaps in regional development. Furthermore, a majority of studies tend to overlook the comparative analysis of spatial coupling when examining decoupling relationships. Incorporating an in-depth analysis of the gravity center's evolution track and spatial coupling can provide supplementary explanations for alterations in regional socioeconomic patterns and the spatial association of various factors.

## 2. Materials and Methods

### 2.1. Study Area

Zhejiang Province is situated along the southeastern coast of China and the southern wing of the Yangtze River Delta (Figure 1). Despite being an economically developed province in China, it is also facing challenges such as limited land resources, high energy consumption, and increasing environmental and resource pressures in the course of social and economic development. According to statistical data, in 2017, Zhejiang Province utilized only 1.5% of the country's arable land to support 4.1% of its population while consuming 4.7% of the country's energy and generating 6.2% of its GDP. As one of the provinces with the smallest land area in China, its mountainous areas account for 74.6%, water surface for 5.1%, and flat areas for 20.3%. There are 32 counties located in mountainous areas or islands.

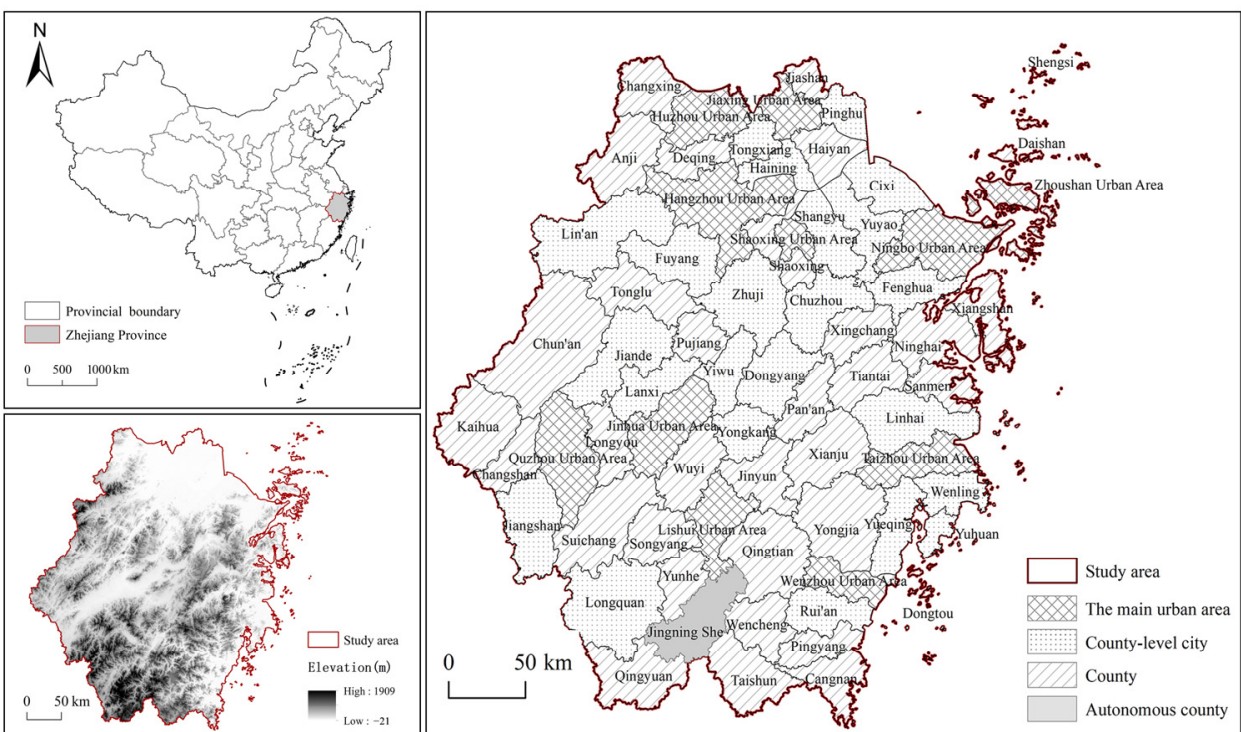

**Figure 1.** Location and study area. (This figure is based on the standard map service website of the Ministry of Natural Resources (http://bzdt.ch.mnr.gov.cn/, accessed on 7 May 2023) with the map approval number of GS(2020)4619).

Considering the lead-lag relationships among economic growth, land expansion, and carbon emissions, it is advisable to conduct decoupling relationship analysis by dividing different time periods [51]. This study includes three comparative periods, namely 2002–2007, 2007–2012, and 2012–2017, which are crucial time nodes in Zhejiang urban development (Figure 2). During the period of 2002–2017, Zhejiang province's urbanization shifted from an economic-driven approach to emphasizing ecological friendliness and resource conservation, with increasing focus on coordinated development across regions. The government has implemented a series of measures in urbanization development, including advancing ecological civilization construction and promoting regional coordinated development to enhance the quality and level of urbanization development.

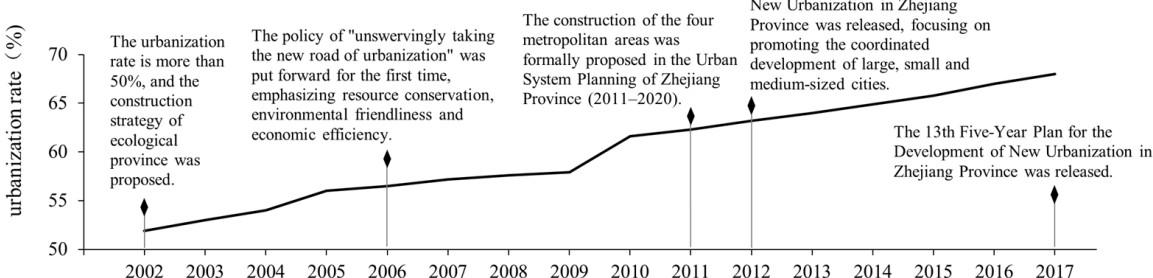

**Figure 2.** Important time nodes of urbanization development in Zhejiang Province.

Due to administrative adjustments during the period of 2002–2017, data collection and analysis were uniformly conducted based on the original administrative units to ensure data comparability. Consequently, 69 research objects were selected, including 23 county-level cities, 34 counties, 11 urban areas, and one autonomous county.

### 2.2. Data Sources

The social and economic data sources used in this study, including regional GDP and built-up area, were gained from the 2002–2017 Statistical Yearbook of Zhejiang and the statistical yearbooks of cities, counties, and districts. We gained the carbon emission data from the China Emission Accounts and Dataset (CEADs) (https://www.ceads.net, accessed on 10 December 2022). In order to ensure comparability, economic data were processed at constant prices in 2002 as the base period.

### 2.3. Decoupling Mechanism Analysis

Decoupling refers to the gradual separation of resource and environmental consumption from the trend of economic growth [52]. Due to the extensive mode of economic development, carbon emissions are rising rapidly and negatively impacting the environment. While implementing measures to reduce carbon emissions may cause a short-term economic slowdown, an orderly transformation to a low-carbon economy can boost economic growth and facilitate the development of eco-civilization in the medium to long term. Land expansion provides new development space for economic growth, which provides financial support for land expansion. However, land resources are limited, and disorderly urbanization hinders sustainable urban development. Hence, improving urban land use efficiency while promoting economic growth is crucial. Achieving a decoupling of economic growth from land sprawl and carbon emissions is conducive to promoting the high-quality development of urban economy and sustainable urban construction (Figure 3).

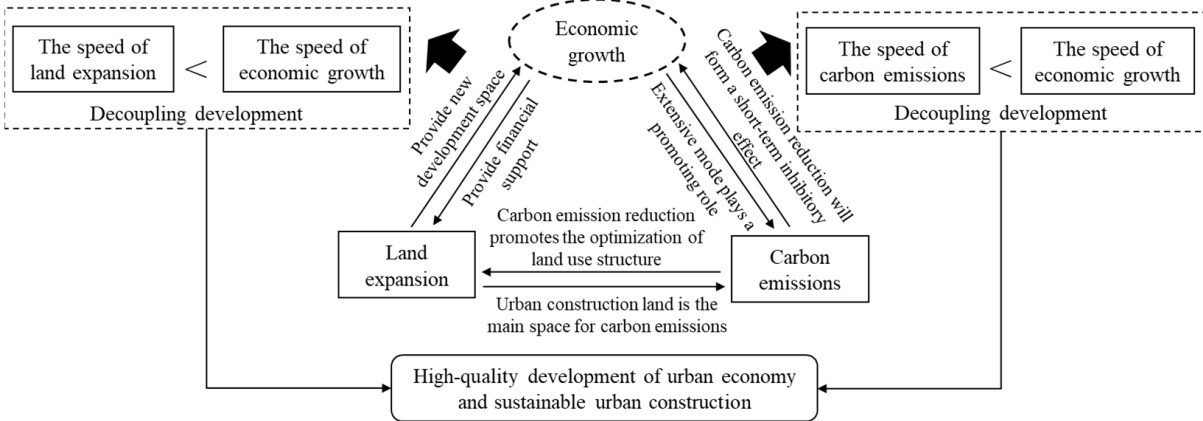

**Figure 3.** Decoupling mechanism examination of economic growth with land expansion and carbon emissions.

### 2.4. Methods

#### 2.4.1. The Gravity Center Model

The gravity center model is for calculating the economic gravity centers, the built-up area, and carbon emissions in Zhejiang Province during the period of 2002–2017. By comparing the distribution of gravity centers in different years, this study explores the evolutionary characteristics and regional differences of economic growth, land expansion, and carbon emissions. The formula used in this model is as follows:

$$\overline{X} = \frac{\sum_{i=1}^{n} P_i X_i}{\sum_{i=1}^{n} P_i} \quad \overline{Y} = \frac{\sum_{i=1}^{n} P_i Y_i}{\sum_{i=1}^{n} P_i} \tag{1}$$

In Equation (1), $\overline{X}$ and $\overline{Y}$ represent the longitude and latitude values of a specific attribute; while $(X_i, Y_i)$ represents the geographical center coordinates of the i-th evaluation unit. $P_i$ represents the attribute value of the *i*-th ($i = 1, 2, 3, \ldots, n$) evaluation unit. In this paper, $P_i$ represents the gross regional product, the built-up area, and carbon emissions.

#### 2.4.2. Spatial Overlap

Spatial overlap measures the spatial coupling degree between the gravity centers of different attributes, which is commonly measured by their spatial distance. The formula for calculating spatial overlap is as follows:

$$S = d_{mn} = \sqrt{(x_m - x_n)^2 + (y_m - y_n)^2} \tag{2}$$

In Equation (2), $S$ represents spatial overlap, $d_{mn}$ represents the spatial distance between the gravity center of attribute m and the gravity center of attribute n. $(x_m, y_m)$ and $(x_n, y_n)$ show the geographical coordinates of the gravity center of attribute *m* and the gravity center of attribute *n* in the same year.

#### 2.4.3. Tapio Decoupling Model

The Tapio decoupling model measures the decoupling relationship by calculating the ratio of the change rate of two correlated variables over a specific period of time. The calculation method is flexible, and the results are not influenced by the research base period, making it widely used in empirical research. The formula is as follows:

$$T_{(B,G)} = \frac{\%\Delta B}{\%\Delta G} = \frac{(B_t - B_{t-1})/B_{t-1}}{(G_t - G_{t-1})/G_{t-1}} \tag{3}$$

$$T_{(C,G)} = \frac{\%\Delta C}{\%\Delta G} = \frac{(C_t - C_{t-1})/C_{t-1}}{(G_t - G_{t-1})/G_{t-1}} \tag{4}$$

In Equations (3) and (4), $T_{(B,G)}$ displays the Tapio decoupling index of economic growth and land expansion, $T_{(C,G)}$ represents the Tapio decoupling index of economic growth and carbon emissions, $\%\Delta G$, $\%\Delta B$, and $\%\Delta C$ represent the change rates of GDP, built-up area, and carbon emissions, respectively. $G$, $B$, and $C$ represent the GDP, the built-up area, and carbon emissions. $T - 1$ and $t$ symbolize the base period and end period.

According to the relationship reflected by the decoupling index, there are 8 types of decoupling status (Figure 4).

The economies of cities in Zhejiang experienced growth from 2002 to 2017; therefore, only "$\Delta G > 0$" was taken into consideration. Zhejiang is leading in terms of economic development, land use intensity, and carbon emission reduction level compared with other areas in China, with relatively small regional differences. We further subdivide the weak decoupling status into weak decoupling I and II based on the general decoupling status classification. To avoid a wide decoupling classification range that would cover the decoupling gap between different cities, T = 0.4 is taken as the dividing line (Table 1).

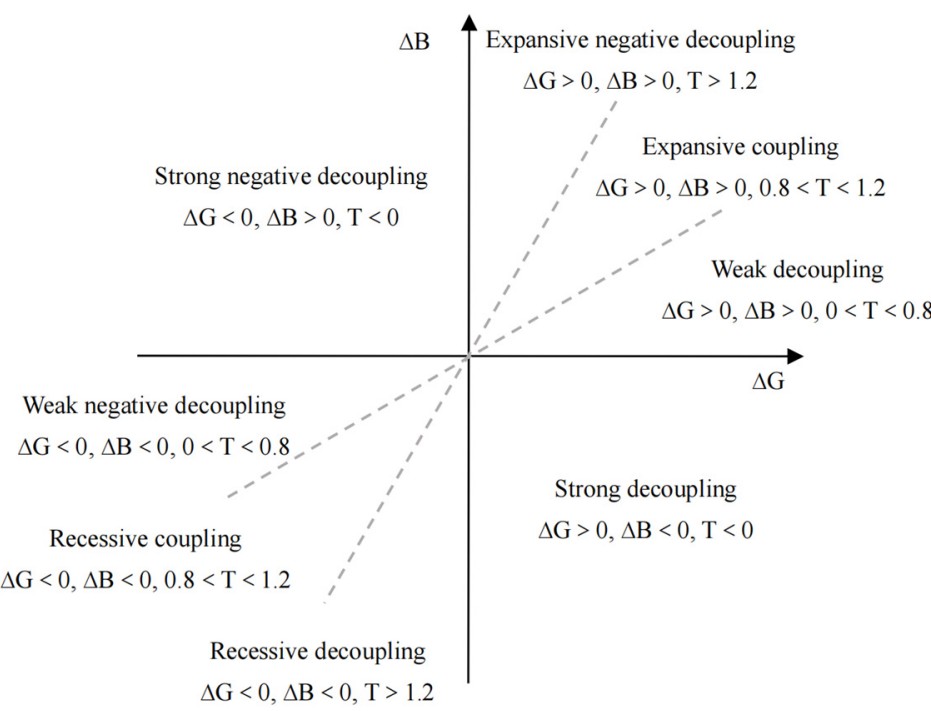

**Figure 4.** The Tapio decoupling index and the decoupling status diagram.

**Table 1.** Refinement of measurement criteria for decoupling status.

| Status | | ΔG | ΔB | Tapio Decoupling Index | Meanings |
|---|---|---|---|---|---|
| Decoupling | Strong decoupling | >0 | <0 | $(-\infty, 0)$ | The economy is growing, and the area of built-up areas is decreasing |
| | Weak decoupling I | >0 | >0 | $(0, 0.4)$ | The economy is growing, and the area of built-up areas is growing very slowly |
| | Weak decoupling II | >0 | >0 | $(0.4, 0.8)$ | The economy is growing, and the area of built-up areas is growing slowly |
| Coupling | Expansive coupling | >0 | >0 | $(0.8, 1.2)$ | The growth rate of the economy and built-up area is close |
| Negative Decoupling | Expansive negative decoupling | >0 | >0 | $(1.2, +\infty)$ | The economy is growing, and the area of built-up areas is increasing significantly |

To better assess the degree of improvement in decoupling status in different stages, we use the change in the decoupling index for dividing the degree of change in the decoupling status into four levels: obvious improvement, slight improvement, slight deterioration, and obvious deterioration (Table 2). The formula is as follows:

$$\Delta = T_{a+1} - T_a \qquad (5)$$

**Table 2.** Standards for improvement in decoupling degrees.

| Type | Δ | \|Δ\| |
|---|---|---|
| Significant improvement | >0 | $(0.4, +\infty)$ |
| Slight improvement | >0 | $(0, 0.4)$ |
| Slight deterioration | <0 | $(0, 0.4)$ |
| Significant deterioration | <0 | $(0.4, +\infty)$ |

In Equations (3) and (4), $T_a$ represents the decoupling index of period a, while $T_{(a+1)}$ symbolizes the decoupling index of period $a + 1$, $|\Delta|$ symbolizes the absolute value of the change in the decoupling index in the two stages. $\Delta > 0$, large $|\Delta|$ value means the improvement is significant. Conversely, $\Delta < 0$, large $|\Delta|$ value represents that the deterioration is significant.

# 3. Results and Analysis

## 3.1. Spatial Coupling Situation of Gravity Centers

### 3.1.1. Contrastive Variation Evaluation of Gravity Centers

Between 2002 and 2017, the economic gravity center, the built-up area gravity center, and the carbon emission gravity center in Zhejiang exhibited movement within a range of 120.49~120.62° E and 29.49~29.63° N, located to the east of the boundary between Zhuji City and Shengzhou City (Figure 5). Relative to the geographical gravity center of Zhejiang, all three gravity centers were located in the northeast, reflecting the long-term uneven geographical distribution of economic scale, land use of the built-up area, and carbon emissions in Zhejiang, with a stronger north–south imbalance than in the east–west direction.

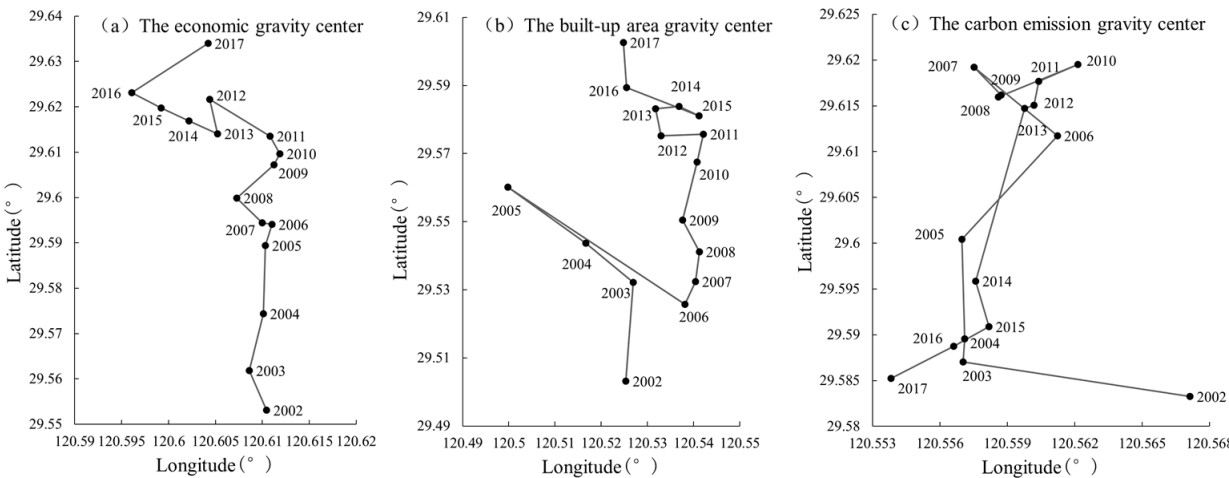

**Figure 5.** The shift of gravity centers of the economy, urban built-up area and carbon emissions in Zhejiang from 2002 to 2017.

Between 2002 and 2017, the economic gravity center of Zhejiang exhibited movement within a range of 120.59~120.62° E and 29.55~29.64° N (Figure 5a), showing a general tendency of moving towards the northwest. Specifically, from 2002 to 2007, the economic gravity center in Zhejiang Province exhibited a noticeable northward movement, driven by the robust economic development momentum of Hangzhou and Ningbo. This period saw a pronounced increase in the primacy of the urban economy, leading to an exacerbation of the north–south economic imbalance within the province. Continuing from 2007 to 2012, the economic gravity center maintained its trajectory towards the northwest, accompanied by an enhanced economic agglomeration effect in the Hangzhou metropolitan area. The economic gravity center experienced significant fluctuations in 2008 due to the tremendous impact of the global financial crisis on the eastern coastal counties and export-oriented cities. Subsequently, from 2012 to 2016, the economic gravity center exhibited a minor southeastward movement before returning to its northwestward path. The proposal and construction of the Jinhua-Yiwu metropolitan area played a crucial role in accelerating economic development in the middle and western regions of Zhejiang. In 2017, the economic gravity center significantly shifted northeastward, primarily driven by the rapid development of the Hangzhou Bay Area, which has emerged as a key driving force behind this movement due to its remarkable growth potential.

Between 2002 and 2017, the built-up area gravity center in Zhejiang exhibited a northwest movement, fluctuating between 120.49~120.55° E and 29.49~29.61° N (Figure 5b). Notably, during 2002–2005, the gravity center shifted significantly northwestwards, driven by the rapid spatial extension of the built-up area in Hangzhou, which expanded from 256 square kilometers to 314.45 square kilometers. However, in 2006, the gravity center shifted notably southeastwards, attributed to the substantial expansion in Ningbo, reaching a size of 215.2 square kilometers. From 2007 to 2011, the gravity center moved towards

the east and north, while coastal counties and cities experienced rapid land expansion. Nevertheless, since 2012, the built-up area gravity center has shown fluctuations and a northwestward trend, with Hangzhou Urban Area and its surrounding regions becoming key areas for land expansion.

Between 2002 and 2017, the carbon emission gravity center of Zhejiang exhibited fluctuations within the geographical coordinates of 120.55~120.57° E and 29.58~29.62° N (Figure 5c), indicating a directional shift towards the northwest, followed by a movement towards the southwest. Specifically, from 2002 to 2007, the carbon emission gravity center moved northwestwards, while the carbon emissions of the majority of counties and cities underwent a rapid growth phase. From 2007 to 2010, the carbon emission gravity center shifted significantly from west to east, as compared to the shift from south to north. The primary reason for this shift was the development of heavy and chemical industries in port areas such as Ningbo Urban Area, Yuyao City, and Cixi City during this stage, leading to the growth of the industrial economy and a significant increase in carbon emissions. From 2010 to 2017, the carbon emission gravity center shifted southwestwards. Notably, the northeastern region of Zhejiang, with high carbon emissions, made significant progress in carbon reduction during this period. Particularly, after Hangzhou became a low-carbon pilot city in China in 2010, various energy-saving and emission-reduction initiatives were implemented in key sectors.

In general, the shifts in economic, built-up area, and carbon emission gravity centers are influenced by a combination of economic growth, industrial development, urban planning, and carbon reduction initiatives in different regions of Zhejiang Province. These factors collectively shape the spatial agglomeration and dispersion patterns of the economy, land use, and carbon emissions, highlighting the significance of regional development strategies and policies to achieve balanced development and sustainable economic growth.

### 3.1.2. Analysis of Gravity Center Coupling

(1)   Spatial coupling of gravity centers of the economy and the built-up area

During the period from 2007 to 2017, the spatial relationship between the economic gravity center and the built-up area gravity center exhibited periodic fluctuations. We divided them into three stages based on the results (Figure 6).

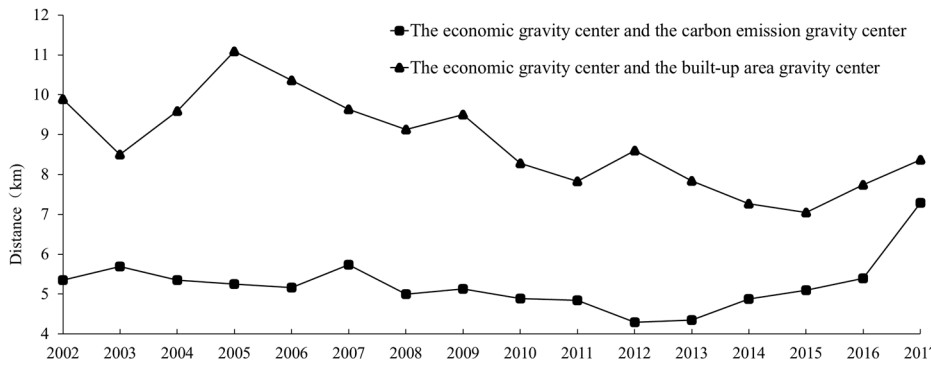

**Figure 6.** The comparison of the distances of the economic gravity center with the built-up area gravity center and carbon emission gravity center in Zhejiang from 2002 to 2017.

From 2002 to 2005, a generally weakened coupling can be seen between the economic and the built-up areas of gravity centers. Specifically, the spatial distance increased from 9.89 km to 11.08 km. The weakening of the coupling was primarily attributed to the imbalance between the demands for land expansion and economic development during this period. The rapid expansion of land has not been effective in stimulating economic development. From 2005 to 2015, the coupling between the economic and the built-up area gravity centers was relatively strengthened. The distance between the two gravity centers decreased from 11.08 km to 7.83 km. The enhancement of coupling was predominantly

influenced by urbanization demands and population migration driven by economic growth during this period. The rapid economic growth of central urban areas, such as Hangzhou Urban Area, attracted a substantial influx of population and concentrated industrial resources, further driving urban land expansion. From 2015 to 2017, the coupling between the economic and built-up areas was relatively weakened. The weakening of the coupling between the two gravity centers was primarily attributed to stringent government restrictions on urban planning and land use, along with a growing focus on environmental protection and sustainable development during this period. Cities with higher economic development levels proactively implemented strict farmland protection measures and promoted intensive land use practices, resulting in a gradual deceleration of urban land expansion.

(2)　Spatial coupling between gravity centers of the economy and carbon emissions

During the period from 2007 to 2017, the spatial distance between the economic and the carbon emission gravity centers exhibited three stages, showing the tendency of first decreasing and then increasing.

From 2002 to 2007, the coupling of the economic and the carbon emission gravity centers remained relatively stable, with the distance between them fluctuating within a narrow range. This period was characterized by various counties and cities being in the rapid urbanization stage, leading to a strong reliance on high-carbon and high-energy-consuming industries for economic growth. Consequently, a notable spatial agglomeration effect was observed between the economic growth gravity center and the carbon emissions gravity center. Subsequently, from 2007 to 2012, the coupling of the economic and carbon emission gravity centers was relatively strengthened, with the spatial distance between them decreasing from 5.74 km to 4.30 km. During this period, regions with rapid economic development, such as Ningbo Urban Area, witnessed the continued expansion of high-carbon and high-energy-consuming industries, which contributed to both high economic output and increased carbon emissions. Finally, from 2012 to 2017, the spatial coupling between the economic and carbon emission gravity centers was relatively weakened, with the spatial distance between them increasing from 4.30 km to 7.29 km. During this period, counties and cities with higher economic development levels proactively adopted industrial transformation and carbon reduction measures driven by technological advancements, policy incentives, and market demands. The growth rate of carbon emissions in the high-carbon emission catchment area around Hangzhou Bay has declined significantly, mainly due to the transfer of heavy industries, optimization and adjustment of industrial structure, and other measures. It is the main reason for the reverse north–south deviation of the trajectories of the carbon emission gravity center and the economic gravity center.

*3.2. Decoupling of Economic Growth with Urban Expansion and Carbon Emissions*

3.2.1. Decoupling of Economic Growth and Urban Expansion

(1)　The Decoupling State between Economic Growth and Land Expansion

At the provincial level, the decoupling of economic growth and land expansion in Zhejiang demonstrates a sign of 'weak decoupling' during the three stages.

At the county level in Zhejiang, there was weak decoupling observed between economic growth and land expansion between 2002 and 2007, with the majority of counties experiencing expansive coupling or expansive negative decoupling states (Table 3 and Figure 7). Notably, eight county units, including Fuyang City and Shaoxing Urban Area, demonstrated expansive negative decoupling, indicating that land expansion did not effectively drive economic growth. In contrast, Zhoushan Urban Area, Shengsi County, and Yueqing City exhibited strong decoupling due to their proactive promotion of the blue ocean economy or the implementation of an improved land quota supply system. From 2007 to 2012, the decoupling of economic growth and land expansion significantly improved, showing an overall positive trend. Compared to the period from 2002 to 2007, there was a significant increase in the number of counties and cities exhibiting weak decoupling level I, while the number of expansive coupling noticeably decreased. Specifically, the

number of weak decoupling level I counties and cities increased by 15, mainly distributed in Taizhou, Jinhua, and other areas. The weak decoupling level II counties and cities were more spatially concentrated, mainly distributed in Huzhou, Jiaxing, Ningbo, and other areas. However, Taishun County, Fenghua County, and Xiangshan County still exhibited a state of expansive coupling or expansive negative decoupling, with relatively low efficiency in intensive land use. From 2012 to 2017, the decoupling of economic growth and land expansion in Zhejiang remained at a weak decoupling level I. However, the number of counties and cities exhibiting expansive coupling and expansive negative decoupling increased somewhat, mainly concentrated in Jiaxing, Shaoxing, and other regions. Notably, Lin'an City, Shaoxing County, Dongtou County, Haining City, Pinghu City, and Jiashan County showed an expansive negative decoupling state, all of which are adjacent to central urban areas. The agglomeration effect of central urban areas has been strengthened, and their leadership role in promoting urbanization construction in surrounding counties and cities is increasingly prominent.

**Table 3.** Number of different types of decoupling of economic growth and land expansion in Zhejiang in different stages.

| Type | Strong Decoupling | Weak Decoupling I | Weak Decoupling II | Expansive Coupling | Expansive Negative Decoupling |
|---|---|---|---|---|---|
| 2002–2007 | 3 | 22 | 23 | 13 | 8 |
| 2007–2012 | 6 | 37 | 23 | 2 | 1 |
| 2012–2017 | - | 39 | 15 | 9 | 6 |

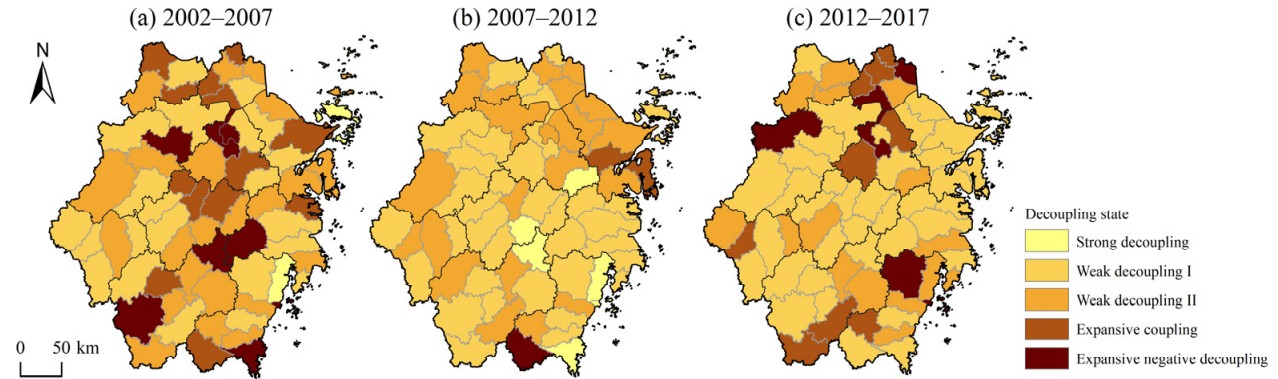

**Figure 7.** Spatial distribution of the decoupling of economic growth and land expansion in Zhejiang during different periods from 2002 to 2017.

The variation in the decoupling degree of economic growth and land expansion between 2007 and 2012, compared to the period from 2002 to 2007, can be categorized into two groups: counties and cities with significant improvement, mainly concentrated in Shaoxing and Jinhua, and counties and cities with slight deterioration, mainly concentrated in Hangzhou, Ningbo, and Wenzhou (Figure 8). During this period, rapid urbanization resulted in outward land expansion in most cities, with a particular concentration of this phenomenon in central cities. During the period of 2012–2017, the decoupling of economic growth and land expansion in Zhejiang saw a significant decrease in the number of counties and cities showing significant improvement, with only six areas experiencing such progress, namely Shaoxing Urban Area, Changxing County, Fenghua City, Xiangshan County, Yiwu City, and Taishun County. The focus of urbanization construction shifted from outward expansion to inward development, resulting in improved decoupling status in central cities such as Hangzhou, Ningbo, Shaoxing, and Wenzhou, where exploitation of construction land stock helped reduce dependence on land expansion during economic growth. However, the decoupling status of the surrounding counties and cities of the

central city deteriorated significantly, indicating challenges in land development efficiency and utilization.

(a) 2007–2012 compared to 2002–2007          (b) 2012–2017 compared with 2007–2012

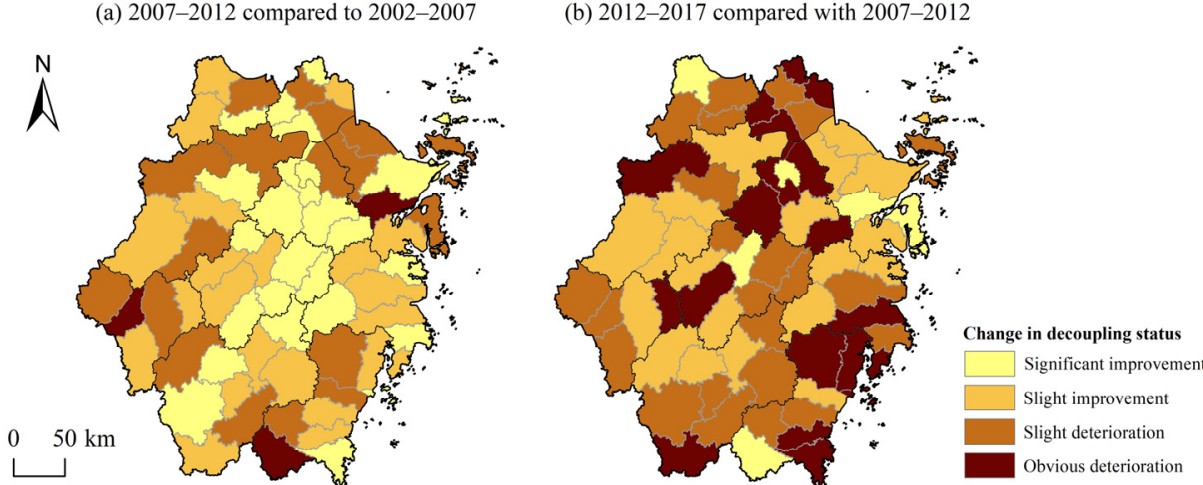

**Figure 8.** The degree of variation in the decoupling of economic growth and land expansion in different periods.

(2)    Division and Transformation Strategy of Urban Types Based on Decoupling Relationship

This study classifies counties and cities based on the decoupling index and the area occupied by built-up areas per unit of GDP and comprehensively evaluates the level of urban land-saving development in each county and city. It takes into the significant deterioration of the decoupling between economic growth and land expansion in many counties and cities from 2012 to 2017, aligning with the stage of deepening new urbanization. The study conducts a comprehensive and in-depth analysis of the economic efficiency of land utilization in different types of cities in Zhejiang Province in the period from 2012 to 2017 and proposes targeted strategies for enhancing the level of efficient and intensive land use. The classification criteria are shown in Table 4, and the specific city types are classified in Figure 9.

**Table 4.** Classification criteria for urban land intensification.

| | Decoupling Index $\in$(0, 0.8) | Decoupling Index $\in$(0.8, 1.2) | Decoupling Index $\in$(1.2, +∞) |
|---|---|---|---|
| The built-up area occupied per unit of GDP $\in$ (0.1, 0.2) | High land consumption, weak decoupling (IV) | High land consumption, expansive coupling (V) | High land consumption, expansive negative decoupling (VI) |
| The built-up area occupied per unit of GDP $\in$ (0, 0.1) | Low land consumption, weak decoupling (I) | Low land consumption, expansive coupling (II) | Low land consumption, expansive negative decoupling (III) |

During 2012–2017, most counties and cities in Zhejiang were concentrated in Region I and Region IV. Region I, with 36 counties and cities, including Shengsi County, Leqing City, and Pingyang County, showed low land consumption per unit of GDP and weak decoupling, indicating effective land resource utilization during economic development. In contrast, Region II, comprising five counties and cities such as Jiashan County, Shangyu City, and Zhuji City, exhibited an expansive coupling state with a land expansion rate comparable to the economic growth rate, necessitating efficient land resource utilization while considering ecological carrying capacity. Region III, consisting of five counties and cities such as Haining City, Pinghu City, and Lin'an City, displayed expansive negative decoupling, with land expansion rate exceeding the economic growth rate, requiring attention to avoid disorderly land expansion and wasteful land use in new developments.

Region IV included 18 counties and cities, such as Taishun County, Shaoxing Urban Area, and Quzhou Urban Area, showing weak decoupling between economic growth and land expansion but high land consumption per unit of GDP. Region V encompassed four counties and cities, namely Qingyuan County, Changshan County, Jiashan Urban Area, and Jingning She Autonomous County, exhibiting an expansive coupling state, urging the promotion of sustainable development, industrial upgrading, and resource-saving and eco-friendly industries. Only Dongtou County was in Region VI, displaying high land consumption per unit of GDP and expansive negative decoupling, posing challenges due to limited land resources while transforming into a district and integrating into the Wenzhou metropolitan area.

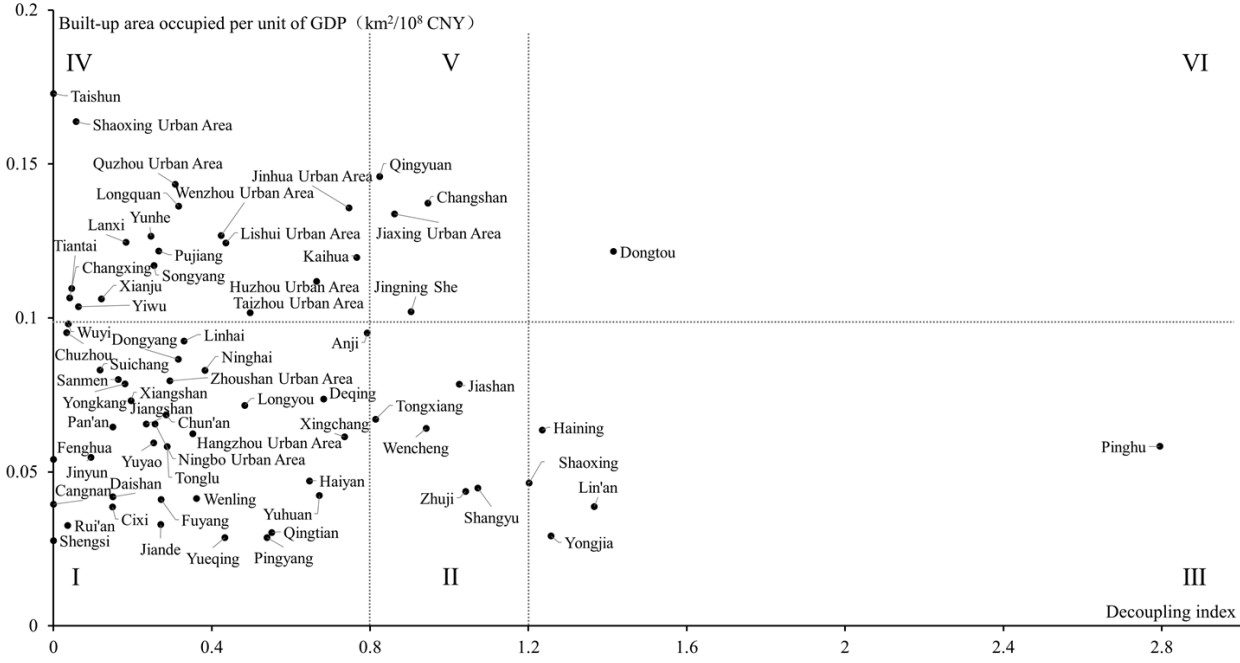

**Figure 9.** A quadrant diagram showing the decoupling index and built-up area per unit GDP of counties and cities in Zhejiang from 2012 to 2017.

In light of the unique patterns of land consumption and decoupling observed in different regions, it is imperative to implement targeted strategies for land control and intensification across the entire province, considering the specific development foundation and industrial patterns of each city.

### 3.2.2. Decoupling of Economic Growth and Carbon Emissions

(1) The Decoupling State between Economic Growth and Carbon Emissions

At the provincial level, Zhejiang has experienced three different periods of decoupling between economic growth and carbon emissions: expansive coupling, weak decoupling II, and strong decoupling.

From 2002 to 2007, most areas at the county level experienced expansive coupling, relying on an extensive mode with high energy consumption and emissions. However, thirteen cities, including Chun'an County, Tonglu County, Tongxiang City, Haining City, and Lishui City, exhibited expansive negative decoupling, primarily in mountainous regions (Table 5 and Figure 10). Meanwhile, weak decoupling II was concentrated on the outskirts of Wenzhou Urban Area, including Anji County and Wencheng County, which implemented the "ecological county" strategy early, promoting green and low-carbon industrial development. From 2007 to 2012, the decoupling status of economic growth and carbon emissions in Zhejiang increased significantly. Weak decoupling levels I and II mainly accounted for 28.99% and 65.22%, respectively. Weak decoupling level I was mainly

observed in mountainous counties and cities in southwestern Zhejiang and the counties and cities surrounding Hangzhou Urban Area, where the carbon emission reduction effect was relatively prominent. However, Jingning She Autonomous Region, Pan'an County, Daishan County, and Shengsi County still experienced expansive coupling or expansive negative decoupling, indicating the need for further promotion of green economic transformation. From 2012 to 2017, strong decoupling became the predominant type of decoupling in the province, accounting for 82.61%. The decoupling index of each county was mainly concentrated in the range of 0–0.4. At this stage, most counties and cities maintained economic growth while effectively controlling total carbon emissions, resulting in a gradual reduction. Weak decoupling level I was scattered across Quzhou and some eastern coastal counties and cities.

**Table 5.** The number of different types of decoupling of economic growth and carbon emissions in Zhejiang during different periods.

| | Strong Decoupling | Weak Decoupling I | Weak Decoupling II | Expansive Coupling | Expansive Negative Decoupling |
|---|---|---|---|---|---|
| 2002–2007 | - | - | 8 | 48 | 13 |
| 2007–2012 | - | 20 | 45 | 3 | 1 |
| 2012–2017 | 57 | 12 | - | - | - |

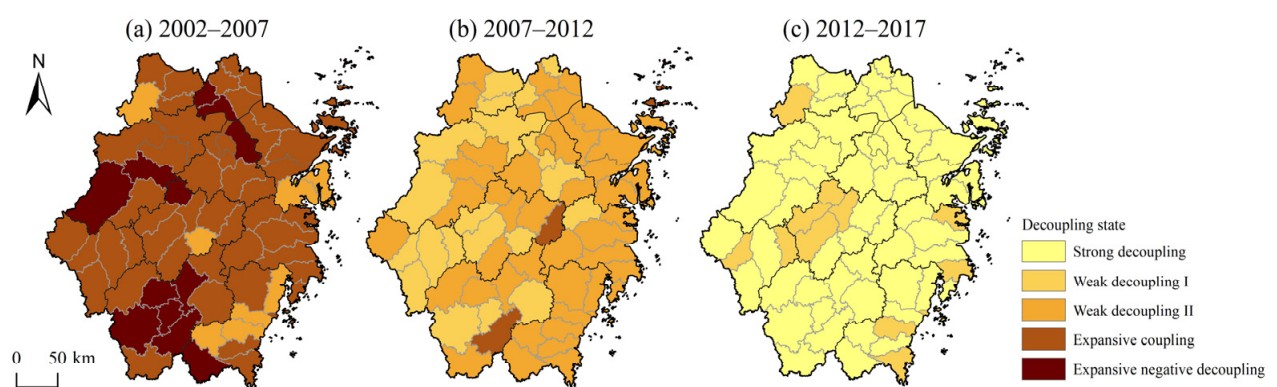

**Figure 10.** The spatial distribution of the decoupling of economic growth and carbon emissions in Zhejiang during different periods.

As the variation of decoupling degree, the degree of economic growth and carbon emissions decoupling status during 2007–2012 improved compared with that in 2002–2007 (Figure 11). However, the degree of improvement in the eastern coastal counties and cities was relatively low. During this period, most counties and cities were in a period of rapid industrial development. Among them, Ningbo vigorously developed port-oriented industries, leading to the rapid growth of energy-intensive and highly polluting industries, which caused great pressure to reduce carbon emissions. Compared to 2007–2012, most counties and cities in Zhejiang significantly improved the decoupling of economic growth and carbon emissions during 2012–2017, with only eight areas showing slight improvement. Fuyang Urban Area, Ningbo Urban Area, Suichang County, and Pan'an County demonstrated notable progress in decoupling by promoting low-carbon industries and innovative development. During the 12th Five-Year Plan, Zhejiang achieved medium to high-speed economic growth while cultivating low-carbon industries such as the information economy and integrating industrialization, informatization, advanced manufacturing, and modern services. In 2012, Ningbo and Wenzhou were designated as the second batch of national low-carbon pilot cities. They achieved effective control of carbon emissions and carbon emission intensity by focusing on the transformation and upgrading of industrial sectors, accelerating the elimination of outdated production capacity, and promoting the development of a circular industrial economy.

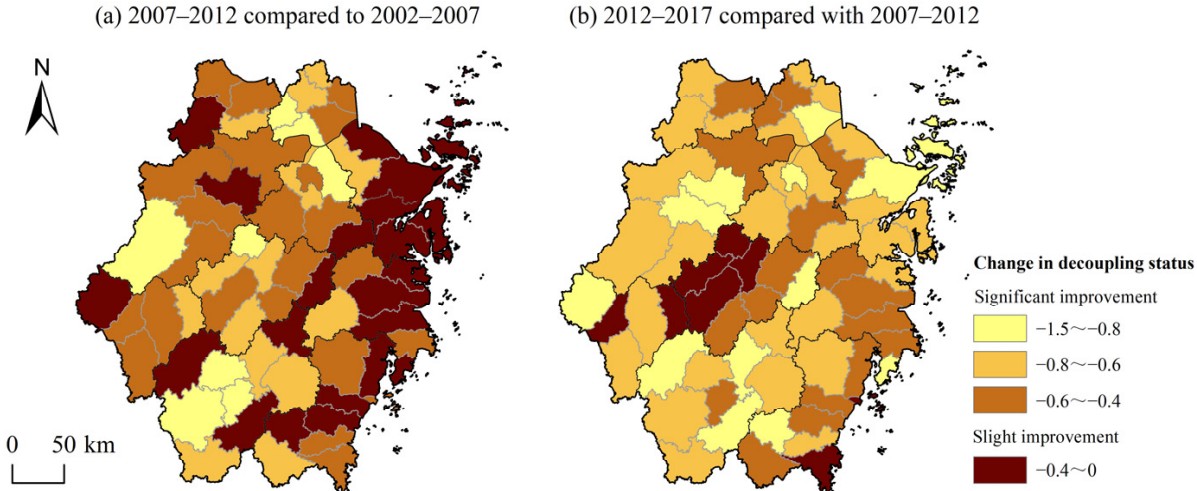

**Figure 11.** The degree of variation in the decoupling of economic growth and carbon emissions in different periods.

(2) Division and Transformation Strategy of Urban Types Based on Decoupling Relationship

This study classifies counties and cities based on the decoupling index and carbon intensity data (carbon dioxide emissions per unit of GDP) and comprehensively evaluates the level of urban low-carbon development in each county and city. It takes into relatively minor regional variations in decoupling status between economic growth and carbon emissions among different cities from 2012 to 2017, aligning with the stage of deepening new urbanization. The study conducts a comprehensive and in-depth analysis of low-carbon economies in different types of cities in Zhejiang Province in the period from 2012 to 2017 and proposes targeted strategies for improving carbon emissions reduction. The classification criteria are shown in Table 6, and the specific city types are classified in Figure 12.

**Table 6.** Classification criteria for urban carbon emission reduction.

| | Decoupling Index $\in (-\infty, 0)$ | Decoupling Index $\in (0, 0.8)$ |
|---|---|---|
| Carbon emission intensity $\in (1, +\infty)$ | High carbon emission intensity, strong decoupling (III) | High carbon emission intensity, weak decoupling (IV) |
| Carbon emission intensity $\in (0, 1)$ | Low carbon emission intensity, strong decoupling (I) | Low carbon emission intensity, weak decoupling (II) |

Most of the counties and cities in Zhejiang are concentrated in Region I and Region III. Specifically, Region I comprises 22 counties and cities, including Hangzhou Urban Area, Wenzhou Urban Area, and Shengsi County. This region exhibits relatively low carbon emission intensity and strong decoupling, achieving a high level of low-carbon urban development. Moving to Region III, which encompasses 35 counties, including Yunhe County, Wencheng County, and Linhai City, we observe high carbon emission intensity but strong decoupling. These counties and cities are generally the primary regions for undertaking the transfer of labor-intensive and high-energy-consuming industries from central urban areas. Notably, Shengsi County stands out for its active promotion of green energy construction on islands. In contrast, Region II includes Rui'an City, Wuling City, and Dongtou County, where carbon emission intensity is relatively low, but the decoupling state remains weak. Meanwhile, Region IV is home to nine counties and cities, namely Changshan County, Sanmen County, and Pujiang County, where carbon emission intensity is high, and negative growth of carbon emissions has not yet been achieved. Notably, Changshan County's carbon reduction progress is relatively slow, and despite its small

economic size, it faces significant pressure due to its heavy industrial structure and large resource consumption.

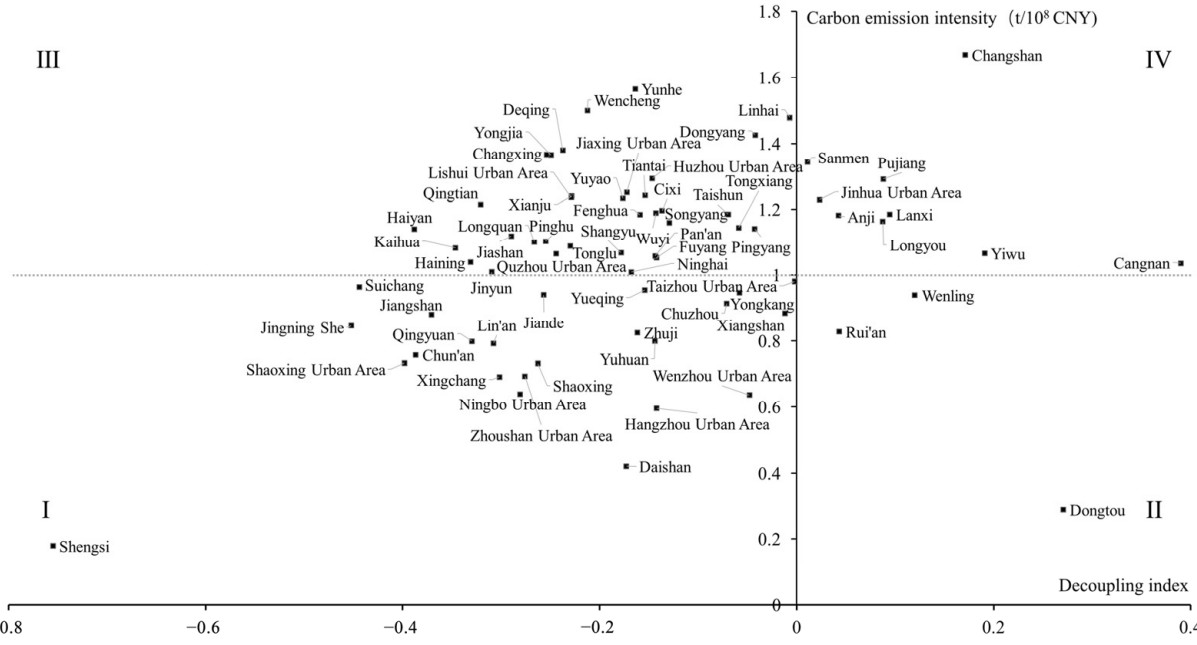

**Figure 12.** A quadrant diagram showing carbon emission intensity and decoupling index of counties and cities in Zhejiang from 2012 to 2017.

Most counties and cities have achieved a strong decoupling between economic growth and carbon emissions. However, there are significant differences in carbon emission intensity among these cities, which are closely related to their dominant industries and energy structure. Therefore, considering tailored carbon reduction measures remains crucial in promoting the maintenance of strong decoupling of carbon emissions in each city.

## 4. Discussions

### 4.1. Characteristics of the Evolution Trajectory of the Gravity Center and Changes in the Decoupling Relationship

This study employed the gravity model to analyze the evolutionary trajectory and spatial coupling of the economic gravity center, land expansion gravity center, and carbon emissions gravity center in Zhejiang Province. It provided insights into the spatial agglomeration and dispersion patterns of the economy, land use, and carbon emissions, as well as their interrelationships and trends. Although Zhejiang is one of the provinces in China with the highest quality of economic development and relatively small regional disparities, spatial coupling analyses revealed that the problem of unbalanced development between counties and cities in Zhejiang still exists. This imbalance is mainly influenced by geographical constraints and provincial spatial development strategies. In general, gravity centers are predominantly located in the developed areas of northeastern coastal Zhejiang. In contrast, mountainous counties face limitations in their economic, land, and industrial urbanization due to topographical constraints. Furthermore, the adjustment of spatial structure strategies, such as metropolitan areas and main functional areas, has also been a significant driving factor for changes in the regional gravity centers of various elements.

In addition, the decoupling model visually identifies the relationship between economic growth, various resource, and environmental elements, providing insights into the sustainability of economic development. This study found that at the county level, the decoupling relationships between economic growth, land expansion, and carbon emissions exhibit distinct spatial heterogeneity and stage-specific variations. At the county level,

decoupling of land expansion and decoupling of carbon emissions exhibit varying spatial heterogeneity and phased corresponding change.

Land expansion decoupling exhibits complex spatial agglomeration, with significant fluctuations concentrated around central cities and surrounding small- and medium-sized cities. Within urban agglomerations, these small- and medium-sized cities generally undergo a transition from weak decoupling to expansive coupling or expansive negative decoupling ("recoupling"). The phenomenon of "recoupling" is also observed in studies conducted at different spatial scales and in other regions [53,54]. However, at the county level, these "recoupling" cities are often situated within urban agglomerations near central urban areas. Despite strong economic connections with the central cities, the process of industry acceptance and urban expansion often results in significant land development with limited economic returns. Therefore, in the process of urban agglomeration development, special attention should be paid to the rational utilization of land resources and the regulation of land development in small- and medium-sized cities. While maximizing the economic radiative effect of central cities, it is essential to consider the spatial balance involving population, resources, and the environment. Simultaneously, efforts should be made to minimize the negative impacts on resources and the environment.

In contrast, the decoupling relationship of economic growth and carbon emissions has transitioned from "expansive coupling, expansive negative decoupling" to "strong decoupling, weak decoupling". Meanwhile, carbon emissions decoupling shows evident spatial convergence due to the integrated carbon reduction policy framework, leading to similar decoupling relationships among cities at different stages. Local governments have implemented carbon reduction plans rooted in natural resources and industrial foundations, promoting demonstrations of diverse types of carbon-reduced cities, which effectively enhance the synergy of carbon reduction efforts. The research findings are consistent with the regional characteristics of decoupling between economic growth and carbon emissions in China, as identified by other scholars. China's eastern coastal regions have seen notable improvements in decoupling carbon emissions [55]. The Yangtze River Delta area stands out with stronger decoupling between GDP and carbon emissions compared to other regions [56]. Within the same timeframe, Zhejiang Province shows minor regional disparities in decoupling status in the Yangtze River Economic Belt [31]. Lu et al. [57] pointed out that the period from 2005 to 2012 represented a pivotal juncture in China's developed regions, characterized by a heightened emphasis on energy conservation, emission reduction, and the role of optimizing industrial structure. Achieving decoupling of economic growth and carbon emissions relies not only on technological advancements but also on adjusting the industrial structure, the transition of a city functional zone, and implementing relevant policies and regulations [58]. Through tailored carbon reduction strategies, the promotion of demonstration projects, and the advancement of industrial transformation and upgrading, cities with different levels of economic development can also achieve relatively balanced decoupling between economic growth and carbon emissions.

The research findings are applicable to other regions in China and developing countries undergoing urbanization transitions, such as India, Brazil, Egypt, etc. These results are particularly valuable for regions entering a new stage of urban agglomeration development, offering important references for achieving more effective land management and carbon emissions reduction as large cities transition into urban clusters.

### 4.2. Policy Recommendations

Considering the diverse decoupling states exhibited by counties and cities, it is crucial to tailor decoupling strategies based on their economic development levels and urban development characteristics, thereby promoting high-quality economic development.

Regarding intensive land use control strategies, counties and cities such as Shengsi County, with low GDP land consumption and weak decoupling, need to further optimize the spatial layout of construction land, promote innovation in land use patterns, tap the value of diversified space to drive land-saving and intensive demonstration projects. For

counties and cities such as Haining City, exhibiting low GDP land consumption but expansive negative decoupling, the emphasis should be on the efficient use of new construction land and full activation of existing construction land during the process of urban-industrial new town development. Counties and cities such as Taishun County, with weak decoupling and high GDP land consumption, need to strengthen land source control, strictly control the scale of new construction land in cities, enhance land resource conservation awareness, and reasonably improve land resource utilization intensity. For counties such as Dongtou County, with expansive negative decoupling and high GDP land consumption, a focus on industry development is crucial. It is recommended that idle and inefficient land be redeveloped to improve the input-output efficiency of land resources while seeking to bridge industrial transformation and land-use transformation in the process of economic transformation.

Regarding carbon reduction strategies, tailored approaches are suggested for various urban contexts as follows. In locales such as Hangzhou Urban Area, which has a low carbon emission intensity and strong decoupling, the strategic focus on low-carbon construction, complemented by the establishment of low-carbon industrial pilot zones, has played a pivotal role as a demonstration. For this kind of densely populated urban center, harnessing the potency of population aggregation becomes paramount, which requires the strong encouragement and promotion of innovative green and low-carbon industries. Turning our gaze towards coastal municipalities, exemplified by Rui'an City, where low carbon emission intensity coexists with weak decoupling, optimal harnessing of marine resources emerges as a linchpin for expanding the horizons of developmental space. The cultivation of distinctive and advantageous sectors, such as eco-friendly fisheries and eco-tourism, stands out as a potent catalyst for stimulating economic advancement. As we delve into mountainous counties and cities characterized by high carbon emission intensity but strong decoupling, such as Yunhe County, enhancing their capability to undertake industrial transfer while leveraging ecological resource advantages is vital for developing low-carbon and pillar industries. Lastly, for regions facing high carbon emission intensity and weak decoupling, such as Changshan County, a focal point lies in reducing energy carbon emission intensity and strengthening source control in high-energy and high-carbon emission sectors. Accelerating the development of green energy construction has become a top priority for the development of these regions.

### 4.3. Limitations and Future Research Direction

Due to the availability of relevant data, analysis for the most recent period has not been conducted, resulting in certain issues of timeliness. Moreover, limitations in traditional statistical yearbook data have led to data lag and insufficient accuracy.

In future research, these limitations can be addressed by utilizing remote sensing data, big data, and other alternative sources to complement and replace the existing data. Additionally, there will be a focus on strengthening empirical research on the factors influencing decoupling changes. This will further elucidate the driving mechanisms behind the variations in the decoupling relationship between economic growth, land expansion, and carbon emissions. Furthermore, the identification of factors influencing these changes in different counties and cities will be a key focus of the research in the future.

### 5. Conclusions

This study starts with regional sustainable development and comprehensively analyzes the spatial and temporal characteristics and coupling trends of economic growth, land expansion, and carbon emissions in different developmental stages of Zhejiang Province. By combining specific development strategies and policy measures, a comprehensive analysis was conducted to evaluate the effectiveness of the transformation towards land intensification and green low-carbon practices in various counties and cities across Zhejiang Province. The conclusions are summarized as follows:

(1) The issue of imbalanced development among counties and cities in Zhejiang Province persists, with incomplete alignment observed between the different gravity centers of economic growth, land expansion, and carbon emissions. This phenomenon is primarily influenced by geographical, environmental conditions and the strategic planning of provincial spatial development.

(2) There is spatial heterogeneity in the decoupling relationship of economic growth with land expansion and carbon emissions, with diverse and fluctuating trends observed over time. The variations in decoupling patterns are closely related to multiple factors, such as the stage of social development, economic growth models, and resource endowment conditions.

(3) The spatial clustering of decoupling status between economic growth and land expansion in Zhejiang Province's counties is relatively complex. Regions experiencing significant fluctuations in decoupling status are mainly concentrated around central cities and their surrounding small- and medium-sized cities, leading to considerable pressure for sustained land expansion decoupling. On the other hand, economic growth and carbon emissions decoupling display evident spatial convergence, with decoupling status consistently improving.

(4) The study proposes strategies to optimize and regulate the decoupling relationship of economic growth with land expansion and carbon emissions, emphasizing the importance of comprehensively considering cities' economic characteristics, environmental conditions, resource endowments, and sustainability when formulating decoupling strategies to achieve strong decoupling.

**Author Contributions:** Conceptualization, Z.L. (Zepan Li) and Z.L. (Zhangwei Lu); Methodology, Z.L. (Zepan Li) and Z.L. (Zhangwei Lu); Formal analysis, Z.L. (Zepan Li); Data curation, Z.L. (Zepan Li); Writing—original draft, Z.L. (Zepan Li), Z.L. (Zhangwei Lu), L.X., Y.S., Q.M., Y.W., Y.C. and B.S.; Writing—review and editing, Z.L. (Zepan Li), Z.L. (Zhangwei Lu) and B.S.; Visualization, Z.L. (Zepan Li); Supervision, Z.L. (Zhangwei Lu), L.X., Y.S., Q.M., Y.W., Y.C. and B.S.; Funding acquisition, Z.L. (Zepan Li) and Z.L. (Zhangwei Lu). All authors have read and agreed to the published version of the manuscript.

**Funding:** This research was funded by the Zhejiang Provincial Natural Science Foundation (No. LY21E080001).

**Data Availability Statement:** Data used in this study are issued by Statistical Yearbook of Zhejiang; China Emission Accounts and Dataset (CEADs). These data can be found here: http://data.cnki.net (accessed on 20 November 2022); https://www.ceads.net (accessed on 10 December 2022).

**Acknowledgments:** The authors gratefully acknowledge the support of the funding.

**Conflicts of Interest:** The authors declare no conflict of interest.

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
