# Peer review of "Examining the Decoupling of Economic Growth with Land Expansion and Carbon Emissions in Zhejiang Province, China"

_land, doi:10.3390/land12081618_

Round 1

Reviewer 1 Report

The paper addresses the problem of the sustainability of economic growth, which is an essential issue that needs to be addressed today. That is, it is necessary to promote economic growth, but this cannot be associated with a deterioration of the environment. To deepen their analysis, the authors analyse the association between economic growth, land use and co2 emissions in a Chinese province. 

The methodology is focused on the calculation of centres of gravity and their evolution over time. The different tools used are not novel, but their combined use gives an interesting insight into the behaviour of the variables. Moreover, it is easily replicable in other works and in other geographical areas and, from my point of view, this adds value to the study. 

The conclusions are consistent with the arguments presented, and the references, tables and figures are appropriate. Regarding these points, I have noted aspects that should be revised/improved on pages 11, 12, 15 and 16. 

Some aspects to be reviewed are indicated below:

- P. 5, eq. 4: revise equation.

- P. 6, figure 4: missing space in "Expansive Coupling".

- P. 9, line 292: missing spaces.

- P. 9, line 31:, change (1) to (2).

- P. 11: the text refers to "slight deterioration" with respect to figure 8, where this option does not appear. Please, clarify.

- P. 12 et seq: it should be indicated how the "decoupling elasticity coefficients" are calculated.

- P. 15, lines 539-543: revise wording.

- P. 16: it should be clarified why negative decoupling index is assimilated with negative growth of carbon emissions.

Reviewer 2 Report

This study starts with the regional sustainable development and comprehensively analyzes the spatial and temporal characteristics and coupling trends of economic growth land expansion, and carbon emissions indifferent developmental stages of Zhejiang Province. The reform of regional economic development strategies and ecological environment policy measures is of great significance. However, the data collected in this article is from 2002 to 2017 and does not have timeliness. It is recommended to use the latest data for analysis.

Minor editing of English language required.

Reviewer 3 Report

RE: Land-2511909 - "Examining the Decoupling of Economic Growth with Land Expansion and

Carbon Emissions in Zhejiang Province, China

Journal: Land

Special Issue: Advances in Land Consolidation and Land Ecology

This paper mainly employs quantitative model for studying the spatial coupling state and decoupling of economic growth with land expansion and carbon emissions in Zhejiang. The topic is interesting, the authors have made an interesting assessment, the subject fits in the general scope of Land, and which is critical for promoting high-quality economic growth and achieving sustainable urban progress.

But in general the expression of results are unclear. This paper fails to engage with the wider readership of Land. Any Description and discussions should be beyond the local case itself, otherwise it cannot attract more international readers. It is recommended that the author make substantial changes before re-review.

The text of the article is written in a difficult-to-understand wording relatively. Needs extensive English editing work in the subsequent modification. In this regard, it should be noted that my knowledge of English as a reviewer is not strong either.

The article language is verbose, not concise enough, the length is too long, the article chapter structure needs to be adjusted, it is suggested to separate the discussion section from Chapter 3 and enrich and deepen the parts.

The expression is not unified, for example, the economic gravity center, the gravity center of built-up area , and the gravity center of carbon emission(Paragraph 1 of 3.1.1 Contrastive variation evaluation of gravity centers) , the economic gravity center, built-up area gravity center, and carbon emission gravity(Part (1) of the 4 conclusions).  Please check the full text and correct any similarities.

Abstract: The part(3) mentioned in the abstract “It is crucial to comprehensively consider...”, which needs to be further reflected in the main text.

Introduction: It lacks reference to international and theoretical literatures on “The connections among economic growth, land expansion, and carbon emissions”. It is suggested that the second paragraph should be sorted out several parts according to the main contents , and relevant international literature should be added.  

Materials and Methods: Second paragraph of 2.1 study area put forward This study includes three comparative periods, namely 2002-2007, 2007-2012, and 2012-2017, however, the following article only covers the situation from 2002 to 2007.

The legend and annotation in Figure 2 are not clear, Also, check the name of the main urban area.

In the Part of 2.4.3, We can't find the %ΔC in formula (4).

The analysis of the results is not concise enough.

The expressions in tables(Table 4, Table 5 and Table6), graphics, and text are not corresponding enough. Such as, Carbon emission intensity in the Table 6, does it represent the same meaning as the carbon emission in the previous paragraph

The discussion is not deep enough and not clear. The discussion needs to be based on the results section of this article and internationalized. For example,based on the result of 3.2 Decoupling of Economic Growth with Urban Expansion and Carbon Emissions, are there similar or different studies in other areas? How adaptive are the results of this paper?

The conclusion needs to be further condensed and looks more like the result at present.

The text of the article is written in a difficult-to-understand wording relatively. Needs extensive English editing work in the subsequent modification. In this regard, it should be noted that my knowledge of English as a reviewer is not strong either.

The article language is verbose, not concise enough, the length is too long, the article chapter structure needs to be adjusted, it is suggested to separate the discussion section from Chapter 3 and enrich and deepen the parts. 

Round 2

Reviewer 3 Report

RE: Land-2511909 - "Examining the Decoupling of Economic Growth with Land Expansion and

Carbon Emissions in Zhejiang Province, China

Journal: Land

Special Issue: Advances in Land Consolidation and Land Ecology

Comments

This paper mainly employs quantitative model for studying the spatial coupling state and decoupling of economic growth with land expansion and carbon emissions in Zhejiang. The authors addressed the previous round comments and made significant improvements to the manuscript. It is recommended publish after minor language and reference check.
